# Binary Search with Distributional Predictions

**Michael Dinitz**[*]
Johns Hopkins University
mdinitz@cs.jhu.edu

**Sungjin Im**[†]
UC Merced
sim3@ucmerced.edu

**Thomas Lavastida**
University of Texas at Dallas
thomas.lavastida@utdallas.edu

**Benjamin Moseley**[‡]
Carnegie Mellon University
moseleyb@andrew.cmu.edu

**Aidin Niaparast**[‡]
Carnegie Mellon University
aniapara@andrew.cmu.edu

**Sergei Vassilvitskii**
Google Research
sergeiv@google.com

## Abstract

Algorithms with (machine-learned) predictions is a powerful framework for combining traditional worst-case algorithms with modern machine learning. However, the vast majority of work in this space assumes that the prediction itself is non-probabilistic, even if it is generated by some stochastic process (such as a machine learning system). This is a poor fit for modern ML, particularly modern neural networks, which naturally generate a *distribution*. We initiate the study of algorithms with *distributional* predictions, where the prediction itself is a distribution. We focus on one of the simplest yet fundamental settings: binary search (or searching a sorted array). This setting has one of the simplest algorithms with a point prediction, but what happens if the prediction is a distribution? We show that this is a richer setting: there are simple distributions where using the classical prediction-based algorithm with any single prediction does poorly. Motivated by this, as our main result, we give an algorithm with query complexity $O(H(p) + \log \eta)$, where $H(p)$ is the entropy of the true distribution $p$ and $\eta$ is the earth mover's distance between $p$ and the predicted distribution $\hat{p}$. This also yields the first *distributionally-robust* algorithm for the classical problem of computing an optimal binary search tree given a distribution over target keys. We complement this with a lower bound showing that this query complexity is essentially optimal (up to constants), and experiments validating the practical usefulness of our algorithm.

## 1 Introduction

Algorithms with predictions, or algorithms with machine-learned advice, has proved to be a useful framework for combining machine learning (which is extremely useful in the usual case but can be quite bad when in the worst case) with traditional worst-case algorithms (which are quite good in the worst-case but do not do as well as we might hope in the usual case). While similar ideas have appeared in many places in the past, the formal study of this setting was pioneered by Lykouris and Vassilvitskii [2021], and was particularly motivated by the practical success of learned index structures [Kraska et al., 2018]. The goal is usually to design an algorithm for some classical and important problem in the setting when we are also provided with some type of "advice" or "prediction" (presumably given by some machine learning algorithm) as to what the instance is like. If the advice is "good" then we want our algorithm to do extremely well, while if the advice is "bad" then we

---

[*]Supported in part by NSF awards 1909111 and 2228995.

[†]Supported in part by NSF awards 1844939, 2121745, and 2423106, and by ONR grant N00014-22-1-2701.

[‡]Supported in part by a Google Research Award, an Infor Research Award, a Carnegie Bosch Junior Faculty Chair, NSF grants CCF-2121744 and CCF-1845146 and ONR Grant N000142212702.

want to inherit the traditional worst-case guarantee. In other words, we want the best of both worlds: good performance in the average case thanks to machine learning, but also robustness and good performance in the worst-case from traditional algorithms.

Consider the basic problem of searching for a key in a sorted array. This problem is the first example considered in the survey of Mitzenmacher and Vassilvitskii [2021], as it is perhaps one of the simplest yet also best-motivated settings. Given a sorted array with $n$ elements and a target key $i$, we can of course do a binary search for $i$ using only $O(\log n)$ comparisons to find its location $\alpha(i)$. But suppose that we additionally receive a *prediction* $\hat{\alpha}(i) \in [n]$ of the location of the target key $i$ in the array, possibly from some machine learning system which attempts to predict the correct location for each key. If the prediction is perfect ($\alpha(i) = \hat{\alpha}(i)$), then it is easy to use this prediction: only one comparison is needed! On the other hand, if the prediction is meaningless, then we can run classical binary search. But what if the prediction is "close"? It turns out that "doubling binary search" from the predicted point can be used to design an algorithm which makes at most $O(\log(|\hat{\alpha}(i) - \alpha(i)|))$ comparisons [Mitzenmacher and Vassilvitskii, 2021]. So if the prediction is very close to the true location then we make very few queries, while if it is very far away then we recover the traditional binary search comparison bound.

The ability to obtain these types of results has led to an explosion of interest in algorithms with predictions; see Section 1.2 for some references. Sometimes the prediction itself is simple, as in the search problem, while sometimes it is quite complex, for instance encompassing a multi-dimensional vector. However, with only a few exceptions such as Diakonikolas et al. [2021], Angelopoulos et al. [2024], which will be discussed in Section 1.2 in detail, all of these papers share an important feature: the prediction itself is non-probabilistic. That is, the prediction is a single (potentially high-dimensional) point (or maybe a small number of such points). While making the setting simpler, this is not a good match with the actual output of most ML systems (particularly modern neural networks), which inherently output a *distribution*. The question we study in this work is how to take advantage of the full richness of the prediction. Of course, we can always turn a distribution into a single point in any number of ways (using a max likelihood estimator (MLE), sampling from the distribution, etc.). But is that always the right thing to do? Or, can we in fact do *better* by taking full advantage of the entire predicted distribution?

## 1.1 Our Results and Contributions

In this paper we initiate the study of algorithms with *distributional predictions*, focusing on the basic search in a sorted array problem described above. In addition to the classic $O(\log n)$ comparisons binary search for an array of size $n$, we recall the "median" or "bisection" algorithm (first described by Knuth [1971] and analyzed by Mehlhorn [1975]) which probes the cell representing the median of the distribution, and recurses appropriately. When the target keys are indeed drawn from the given distribution, the expected query complexity (i.e. number of comparisons between elements in the array and the target) is bounded by $H(p)+1$, where $H(p)$ is the entropy of the distribution [Mehlhorn, 1975]. (Note that when the distribution is uniform over $n$ elements, this recovers the $O(\log n)$ binary search bound). This is in fact essentially optimal: it is known that every algorithm requires at least $H(p)/3$ queries in expectation when target keys are drawn from $p$ [Mehlhorn, 1975].

On the other hand, it is easy to see that if target keys are *not* drawn from $p$, then this algorithm can be arbitrarily bad: it can easily be made to use $\Omega(n)$ comparisons in expectation. So our main question is the following: how can we best utilize a prediction $\hat{p}$ which is *not* the true distribution $p$? Can we recover the near-optimality of the median algorithm without being subject to its worst-case performance?

**Reduction to point distributions.** We first show in Section 2.1 that the obvious approach, of reducing $\hat{p}$ to a point prediction (whether by sampling, using a max-likelihood prediction, or some other method) and then using previous algorithms, is a bad idea that can lead to poor worst-case performance. In addition to ruling out a natural class of algorithms, this gives additional motivation to our study of distributional predictions: as discussed, essentially all previous work studies the case in which the prediction is a single point (in this case, location); yet, most machine-learning systems will actually output a distribution. So our lower bound implies that any of these traditional point-based algorithms, no matter how good the bound obtained compared to their prediction, must

suffer fundamentally poor performance in the real world where target keys actually come from a distribution.

**Main algorithm.** We then give our main result in Section 3: an algorithm which interleaves phases of the "median" algorithm and classical binary search to obtain a query complexity of $O(H(p) + \log \eta)$, where $\eta$ is the earth mover's distance (EMD, also known as the Wasserstein $W_1$ metric) between $p$ and $\hat{p}$, See Section 3 for precise details. Note that $H(p) \leq O(\log n)$ and $\log \eta \leq \log n$. So if our prediction $\hat{p}$ is close to $p$, then our algorithm has performance essentially equal to the best possible bound $H(p)$. On the other hand, if our prediction $\hat{p}$ is far from $p$ (so provides essentially no information), we do not suffer the poor performance of naively believing in $\hat{p}$ and running the median algorithm on it, instead recovering a bound of $O(H(p) + \log \eta) = O(\log n)$.

While there are many notions of "distance" between distributions, EMD is a natural one in our setting. Many other notions of distance, like $\ell_1$, do not take the geometry of the line into account. For example, consider some distribution $p$ over $[n]$, and let $p'$ be the distribution obtained from $p$ by moving $\gamma/2$ probability mass from 1 to 2, and let $p''$ be the distribution obtained from $p$ by moving $\gamma/2$ probability mass from 1 to $n$. Then the $\ell_1$ distance between $p$ and $p'$ is $\gamma$, and so is the $\ell_1$ distance between $p$ and $p''$. Yet clearly $p'$ is a "more accurate" prediction for $p$. The earth mover's distance recognizes this fact, and so is a more appropriate measure than $\ell_1$. Similarly, popular measures such as KL-divergence (which are not technically metrics, but do give a notion of distance) suffer the same flaws as $\ell_1$ while also being extraordinarily sensitive to mismatches in the support (the KL-divergence can be infinite if the supports of the two distributions do not agree).

**Distributional Robustness of Optimal Binary Search Trees.** We have so far discussed a distributional prediction setting where a target key arrives with a predicted distribution $\hat{p}$ of its location. This is a strict generalization of Mitzenmacher and Vassilvitskii [2021], where the prediction is a single location in the array. In their model, the location is error prone and in ours the distribution over locations is error prone. Our goal is to construct an effective search strategy given the target and the prediction.

But there is another related setting: there is a single (unknown) distribution over target keys, and we are given a (possibly erroneous) prediction of this distribution and are asked to design a search algorithm with minimum expected lookup time when target keys are drawn from the true distribution. In other words, instead of each target key coming with a predicted distribution $\hat{p}$ over *locations* in $[n]$ and the true location being drawn from some true distribution $p$, we are given ahead of time a predicted distribution $\hat{p}$ over *target keys* $[n]$, and are asked to design a lookup algorithm for inputs that use this distribution. But then these target keys in the input are actually drawn from $p$ rather than $\hat{p}$, and do not come with target-specific predictions.

Since any comparison-based search algorithm is equivalent to a particular binary search tree, if $\hat{p} = p$ then this is precisely the classical problem of computing an *optimal binary search tree* [Mehlhorn, 1975]. So we can interpret our results as providing distributionally-robust optimal BSTs: given $\hat{p}$, we can efficiently compute a BST where the expected lookup time under the true (but unknown) query distribution $p$ is at most $O(H(p) + \log \eta)$. Surprisingly, given the classical nature of computing optimal (or near-optimal) BSTs, this simple question of "what if my distribution is incorrect?" has not been considered in the data structures and algorithms literature.

**Worst case lower bound.** We complement our algorithmic development with a lower bound in Section 3.2, proving that no algorithm can use fewer than $\Omega(\log \eta)$ queries in the worst case. Since $\Omega(H(p))$ is a known lower bound as well even if $p$ is known perfectly [Mehlhorn, 1975], this implies that our algorithm is asymptotically tight. So if we measure accuracy of the prediction via EMD, no algorithm can make asymptotically better use of a distributional prediction.

**Portfolios of predictions.** There has been recent interest in the study of algorithm with *multiple* predictions, sometimes called prediction *portfolios*. See, for example, [Balcan et al., 2021, Dinitz et al., 2022, Anand et al., 2022, Kevi and Nguyen, 2023]. The goal is usually to do as well as the *best* of the predictions in the portfolio, with the difficulty being that we do not know *a priori* which of these predictions is best. We extend our main algorithm to this setting in Section 4, showing that it is possible to use *multiple* distributional predictions effectively.

**Experiments.** Finally, in Section 5 we give empirical evidence of the efficacy of the algorithm we propose. We first use synthetic data to demonstrate the effect of distribution error on the performance of the algorithm. We then evaluate it on a number of real world datasets.

## 1.2 Other Related Work

Machine learning augmented algorithms have found applications in various areas — for example, online algorithms [Lykouris and Vassilvitskii, 2021, Purohit et al., 2018], combinatorial algorithms [Dinitz et al., 2021, Davies et al., 2023], differential privacy [Amin et al., 2022], data structures [Lin et al., 2022, Vaidya et al., 2021, McCauley et al., 2024, Mccauley et al., 2024] and mechanism design [Agrawal et al., 2022], to name a few. In particular, online algorithms have been extensively studied with ML advice for various problems, such as online caching [Lykouris and Vassilvitskii, 2021], ski-rental [Purohit et al., 2018], scheduling [Lattanzi et al., 2020], knapsack [Im et al., 2021], set cover [Bamas et al., 2020], and more [Lindermayr and Megow, 2022]. Due to the vast literature, we only provide a few samples.

Particularly relevant to our setting is the work of Lin et al. [2022], which initiated the study of predictions for binary search trees. This work investigates how to improve a treap's guarantees when item frequencies follow distributions such as the Zipfian distribution.

As discussed earlier, in ML augmented algorithms, predictions are typically given in the form of specific values, rather than distributions. Here, we discuss a few exceptions. The work by Diakonikolas et al. [2021] studies the ski-rental problem and prophet inequalities with access to i.i.d. samples from an unknown distribution. Their focus lies on the sample complexity and not on the correctness of the distribution. Indeed, they obtain robustness by combining their consistent algorithm and the best worst-case algorithm in a black-box manner. In contrast, we assume full access to a distributional prediction and develop new ideas to obtain a tight trade-off between consistency and robustness, in conjunction with natural error measures involving entropy and the earth mover's distance, which take the accuracy of the distributional prediction into account.

More broadly the question of how to improve performance of problems where either full instances, or some model parameters come from a known distribution is well studied under the rubric of two-stage stochastic optimization [Swamy and Shmoys, 2006, Ahmed, 2010], with techniques like Sample Average Approximation (SAA) [Kim et al., 2015] having a rich history. Similar to our setting, one can look at the robust setting, where the distribution available to the algorithm is different from the true distribution that examples are drawn from, see e.g., [Bertsimas and Goyal, 2010, Dütting and Kesselheim, 2019, Bertsimas et al., 2022, Besbes et al., 2022]. Typically in these cases one assumes a bound on the difference between the two distributions, explicitly choosing what to hedge against, and then derives optimal strategies. In contrast, in this work we aim to find a smooth trade-off on the performance of the algorithm as a function of the distance between the two distributions, coupled with an upper bound on worst-case performance.

Finally, the very recent work of Angelopoulos et al. [2024] is one of the few that consider distributional predictions. It shows an optimal tradeoff between consistency and robustness for a scheduling problem. However, their solution space explored is considerably more limited than ours, essentially consisting of geometric sequences with a multiplicative ratio of 2, each characterized by its starting point. Further, their bound analysis is restricted to cases where the error is sufficiently small. In contrast, our work demonstrates how a binary search algorithm can compare generally to earth mover's distance (EMD) and a lower bound on the optimum for any predicted distribution. As a result, we develop novel algorithmic solutions that build upon a close connection to EMD.

## 2 Preliminaries

We now formally describe our problem and setting. Let $a_1 < \cdots < a_n$ be a set of $n$ keys, and $p = (p_1, \ldots, p_n)$ be a probability distribution over the keys. Our goal is to develop a search strategy (or search algorithm), which takes a target key $a$ as input, and finds a position $i$ such that $a_i = a$. We aim to find search strategies with low expected search cost when the target key $a$ is sampled according to the distribution $p$. In our analysis we will consider the number of comparisons, also known as the *query complexity*, as the main metric of study. This metric captures the information theoretic complexity of the problem, and ignores computational overhead. Formally, let the search

cost $C(a_i)$ of finding $a_i$ be the number of comparisons done by the algorithm when the target key is $a_i$. The expected search cost is then $\sum_{i=1}^{n} p_i C(a_i)$.

To aid in this goal, we are given a prediction, $\hat{p} = (\hat{p}_1, \ldots, \hat{p}_n)$, of $p$. To account for the fact that the predicted distribution may be incorrect, we let $\eta$ denote the earth mover's distance (EMD) between $p$ and $\hat{p}$. The EMD between two distributions $P$ and $Q$ is the solution to the optimal transport problem between them, or more formally, it is $\inf_{\gamma \sim \Pi(P,Q)} \mathbb{E}_{(x,y) \sim \gamma}[d(x,y)]$, where $\Pi(P,Q)$ is the set of joint distributions with marginals $P$ and $Q$.

Finally, for a distribution $p$, we let $H(p) = -\sum_{i=1}^{n} p_i \log(p_i)$ be the entropy of $p$. Throughout the paper, all logarithms are in base 2.

Given the breadth of work on point predictions, it is tempting to try and reduce the distributional prediction problem to point predictions. We show that this approach does not lead to good results.

## 2.1 Point Predictions from Distributions

Given prediction $\hat{p}$ of $p$, suppose the algorithm first computes some point $\hat{\alpha}$ from $\hat{p}$ and then uses the doubling binary search algorithm from Mitzenmacher and Vassilvitskii [2021] with prediction $\hat{\alpha}$. This will have expected running time of $O(\log(|\hat{\alpha} - \alpha|))$, where $\alpha$ is the true location of the key. A natural question is whether there is some $\hat{\alpha}$ so that $O(\log(|\hat{\alpha} - \alpha|))$ is comparable to $O(H(p) + \log \eta)$.

Unfortunately, this is not possible, necessitating our more involved algorithm and analysis. Let $p$ be the distribution on two atoms, with $p_{n/4} = 1/2$ and $p_{3n/4} = 1/2$, and let $\hat{p} = p$. Clearly $\eta = 0$, and $H(p) = 1$, so any competitive algorithm must terminate after a constant number of comparisons in expectation. On the other hand, consider some $\hat{\alpha} \in [n]$. If $\hat{\alpha} \leq n/2$, then since $\alpha = 3n/4$ with probability $1/2$ we have that $\mathbb{E}[\log(|\hat{\alpha} - \alpha|)] \geq \frac{1}{2} \log(n/4)) = \Omega(\log n)$. Similarly, if $\hat{\alpha} \geq n/2$, then since $\alpha = n/4$ with probability $1/2$ we have that $\mathbb{E}[\log(|\hat{\alpha} - \alpha|)] \geq \Omega(\log n)$. Hence converting $\hat{p}$ to a point prediction and then using the algorithm of Mitzenmacher and Vassilvitskii [2021] as a black box is doomed to failure.

# 3 Algorithm

To develop our robust approach, recall the two baseline algorithms—traditional binary search with an $O(\log n)$ running time and the algorithm that recurses on the median element of the distribution, with an $O(H(p))$ running time (assuming it has access to the true distribution $p$).

In our algorithm we interleave these two approaches to get the best of both worlds. Let $a \in \{a_1, \ldots, a_n\}$ be the target key. We proceed recursively, keeping track of an active search range $[\ell, r]$ (if $a = a_i$, we always have $i \in [\ell, r]$). Initially, we start with $\ell = 1$ and $r = n$. The algorithm proceeds in iterations. Each iteration $i$ for $i = 0, 1, \ldots$ has two phases

- **Bisection.** Divide the search range in half based on the predicted probabilities $\hat{p}$. Formally, find an index $k$, $\ell \leq k \leq r$ such that $\sum_{j=\ell}^{k-1} \hat{p}_j \leq \frac{1}{2} S$ and $\sum_{j=k+1}^{r} \hat{p}_j \leq \frac{1}{2} S$, where $S = \sum_{j=\ell}^{r} \hat{p}_j$. Compare $a$ to $a_k$. If they are equal, return $k$. Otherwise, based on the result of the comparison, continue the search on the ranges $[\ell, k-1]$ or $[k+1, r]$.

  Continue this process for $2^i$ steps, and if $a$ is not found, begin the second phase.

- **Binary Search at the Endpoints.** Let $[\ell, r]$ be the current search range. Set $d = \min(2^{2^i}, r - \ell)$. Check if $a$ is in the range $[\ell, \ell + d]$ or $[r - d, r]$ (by comparing $a$ to $a_{\ell+d}$ and $a_{r-d}$). If $a$ is in one of these ranges, say $[\ell, \ell + d]$, do a regular binary search (by choosing the middle point of the range each time) on the range $[\ell, \ell + d]$, until $a$ is found. Otherwise, start the next iteration with the new search range $[\ell + d + 1, r - d - 1]$.

The algorithm continues until $a$ is found.

## 3.1 Analysis

The goal is to show the following theorem with respect to the algorithm.

**Theorem 3.1.** *The expected query complexity of the described algorithm is at most* $4H(p) + 8\max(\log(\eta)+2,1)+8 = O(H(p)+\max(\log(\eta),0))$.[4]

Before formally proving the theorem, we give key intuition about the analysis. In iteration $k$, the Bisection phase is continued for $2^k$ steps. In each step, one comparison is made, which makes the cost of this phase $2^k$.

Consider the Binary Search at the Endpoints phase. Two comparisons are made during the phase unless $a \in [\ell, \ell+d]$ or $a \in [r-d, r]$. In those cases, we run a traditional binary search on an interval of length $d+1$, whose cost is $\log d = \log 2^{2^k} = 2^k$.

For each key $a_i$, it takes at most $\log(\frac{1}{p_i})+1$ iterations of the Bisection phase to get to a search range that has a predicted probability mass of at most $p_i/2$. We can charge the total cost of the iterations up to this point to the term $p_i \log(\frac{1}{p_i})$ in $H(p)$.

Either we find $a_i$ earlier, in which case the total cost of the iterations can be charged to $H(p)$, or there is an at least $p_i/2$ mass that was predicted to lie outside the interval, allowing us to lower bound $\eta$. We make this argument formal below.

*Proof of Theorem 3.1.* With probability $p_i$, the target key $a$ that we are looking for is $a_i$. The goal is to bound the expected cost of the algorithm, which is $\sum_{i=1}^{n} p_i C(a_i)$, where $C(a_i)$ is the cost of the algorithm when $a = a_i$. Let $k_i$ be the first iteration at which $a$ is found, assuming $a = a_i$. As mentioned earlier, the total cost of the first phase of the iterations 0 to $k_i$ is $\sum_{j=0}^{k_i} 2^j < 2^{k_i+1}$. Also, the cost of the second phase in each iteration before $k_i$ is 2, and in iteration $k_i$ is at most $2^{k_i}$. So the total cost of the algorithm for iterations 0 to $k_i$ is at most $3 \cdot 2^{k_i} + 2k_i \leq 4 \cdot 2^{k_i}$. We partition the keys based on $k_i$ into two sets, and bound the cost of each set separately. Let $I_1 := \{i : k_i \leq \log(\log(4/p_i))\}$ and $I_2 := \{i : k_i > \log(\log(4/p_i))\}$.

First, we bound the cost of indices in $I_1$ by a constant factor of $H(p)$:

$$\sum_{i \in I_1} p_i C(a_i) \leq 4 \sum_{i \in I_1} p_i 2^{k_i} \leq 4 \sum_{i \in I_1} p_i \log(4/p_i) \leq 4H(p) + 8.$$

Now we bound the cost of the indices in $I_2$ by a constant factor of $\log(\eta)$. Let $i \in I_2$. We know that during iteration $j$ of searching for $a_i$, in the Bisection phase, the predicted probability mass in the search range decreases by a factor of at least $2^{2^j}$. Therefore the predicted probability mass in the search range $[\ell, r]$ at the end of the first phase in iteration $k_i - 1$ is at most

$$\frac{1}{\prod_{j=0}^{k_i-1} 2^{2^j}} = \frac{1}{2^{2^{k_i}-1}} = \frac{2}{2^{2^{k_i}}} \leq \frac{2}{4/p_i} = \frac{p_i}{2},$$

where the inequality holds because $i \in I_2$. So $\sum_{j=\ell}^{r} \hat{p}_j \leq p_i/2$. Let $D_i := \min(i - \ell, r - i)$. In the transportation problem corresponding to the earth mover's distance between $p$ and $\hat{p}$, a probability mass of at least $p_i/2$ needs to be moved from point $i$ to the outside of the interval $[\ell, r]$. The cost of this movement in the objective function of the transportation problem is at least $D_i \cdot p_i/2$. Therefore we have $\eta \geq \sum_{i \in I_2} D_i \cdot p_i/2$. In the Binary Search at the Endpoints phase of iteration $k$, we probe indices within distance $d = 2^{2^k}$ around the two endpoints of the search range. Since $a_i$ is not found before iteration $k_i$, we conclude that $2^{2^{k_i-1}} < D_i$, which means that $2^{k_i} \leq 2\log(D_i)$. Let $p(I_2) := \sum_{i \in I_2} p_i$. We have

$$\sum_{i \in I_2} p_i C(a_i) \leq 4 \sum_{i \in I_2} p_i 2^{k_i} \leq 8 \sum_{i \in I_2} p_i \log(D_i) = 8 \left( \sum_{i \in I_2} p_i \log(D_i) + (1 - p(I_2))\log(1) \right).$$

By concavity of the $\log(\cdot)$ function and Jensen's inequality we have

$$8 \left( \sum_{i \in I_2} p_i \log(D_i) + (1 - p(I_2))\log(1) \right) \leq 8 \log \left( \sum_{i \in I_2} p_i D_i + (1 - p(I_2)) \right) \leq 8 \max(\log(\eta)+2, 1).$$

---

[4]To account for the case where $\eta \in [0, 1)$ where $\log(\eta) < 0$, we impose a bound by taking the maximum of $\log(\eta)$ and 0.

The last inequality follows from the fact that if $\sum_{i \in I_2} p_i D_i \leq 1$ we have $\sum_{i \in I_2} p_i D_i + (1 - p(I_2)) \leq 2$, and otherwise we have

$$\log\left(\sum_{i \in I_2} p_i D_i + (1 - p(I_2))\right) \leq \log\left(\sum_{i \in I_2} p_i D_i\right) + 1 \leq \log(2\eta) + 1 = \log(\eta) + 2. \qquad \square$$

## 3.2 Lower Bound

It is well known that there is a lower bound of $\Omega(H(p))$ on the expected query complexity for binary search [Mehlhorn, 1975]. We now show that there is a lower bound of $\Omega(\log \eta)$ on the expected query complexity as well, even on instances with $H(p) = 0$. This shows that there must fundamentally be a $\log \eta$ dependence on the earth mover's distance, even for instances where it is not absorbed by the dependence on the entropy.

**Theorem 3.2.** *For any $\eta \in [n]$, any comparison-based (deterministic or randomized) algorithm must make $\Omega(\log \eta)$ queries on some instance where $H(p) = 0$ and the earth mover's distance between $p$ and $\hat{p}$ is $O(\eta)$.*

*Proof.* Thanks to Yao's principle, it suffices to give a distribution over instances of this problem and argue that any deterministic algorithm has a large expectation over this distribution. Let the set of keys be $[n]$. We present a family of problem instances $I_1, \ldots, I_\eta$, where each instance can happen with probability $\frac{1}{\eta}$. In instance $I_i$, the true access distribution is a singleton on location $i$, i.e., in $I_i$ we have $p_i = 1$ and $p_j = 0$ for each $j \in [n] \setminus \{i\}$. In all the problem instances $I_1, \ldots, I_\eta$, the prediction is the uniform distribution over $[\eta]$. Note that for each instance $I_i$, we have $H(p) = 0$. Also, the earth mover's distance between $\hat{p}$ and $p$ is at most $\eta$.

Our claim is that any deterministic algorithm has an expected cost of $\Omega(\log \eta)$ over this distribution. To see this, note that the expected cost of any deterministic algorithm over this distribution of instances exactly equals the cost of that algorithm on an instance $I^*$ where the true access distribution is uniform over $[\eta]$. Now by the lower bound of Mehlhorn [1975], the cost of any deterministic comparison-based algorithm on $I^*$ is $\Omega(H(p^*))$, where $p^*$ is the uniform distribution over $[\eta]$. To conclude the proof, note that $H(p^*) = \Omega(\log \eta)$. $\qquad \square$

Combining Theorem 3.2 with the $\Omega(H(p))$ lower bound due to Mehlhorn [1975] results in the following worst-case lower bound, asymptotically matching Theorem 3.1.

**Corollary 3.3.** *Any comparison-based algorithm for binary search with distributional predictions has worst-case expected query complexity $\Omega(H(p) + \log(\eta))$.*

# 4 A Portfolio of Predictions

In the previous section we showed an algorithm that is optimal given a single distributional prediction. Here we extend this result to the setting where there are $m$ different distributions given as a prediction. That is, for $k \in \{1, 2, \ldots, m\}$, there are predictions $\hat{p}_k = (\hat{p}_{1,k}, \ldots, \hat{p}_{n,k})$ of $p$ given. Let $\eta_k$ be the earth mover's distance between $\hat{p}_k$ and $p$. The goal is to design an algorithm competitive with *single best* distribution $\hat{p}_k$. That is, comparable to $\min_{k \in [m]} \log \eta_k$ and $H(p)$.

## 4.1 Algorithm for Multiple Predictions

We proceed in a similar manner, alternating the two phases. However, we change the algorithm so that in the first phase the algorithm performs a binary search on the medians of each distribution. The goal of this is to ensure that each distribution has its probability mass drop by at least half in each step.

As before, the initial search range is $[1, n]$. The algorithm proceeds in iterations. For $i = 0, 1, \ldots$, iteration $i$ has two phases.

- **Bisection.** Let $[\ell, r]$ be the current search range. Let $S_k = \sum_{j=\ell}^{r} \hat{p}_{j,k}$ be the remaining probability mass in the $k$'th prediction.

Let $t_k$ be such that $\sum_{j=\ell}^{t_k-1} \hat{p}_{j,k} \leq \frac{1}{2} S_k$ and $\sum_{j=t_k+1}^{r} \hat{p}_{j,k} \leq \frac{1}{2} S_k$. That is, $t_k$ is the median of the $k$'th distribution. Sort the indices $k \in [m]$ so that $t_1 \leq t_2 \ldots \leq t_m$. For convenience, let $t_0 = \ell$ and $t_{m+1} = r$. Perform a binary search on $a_{t_0}, a_{t_1}, a_{t_2}, \ldots a_{t_{m+1}}$ to find the interval where $a \in (a_{t_j}, a_{t_{j+1}})$ for some $j \in \{0, 1, 2, \ldots m\}$. The new search range is $[t_j + 1, t_{j+1} - 1]$.

Continue this for $2^i$ steps, and if $a$ is not found, begin the second phase described below.

- **Binary Search at the Endpoints.** Let $[\ell, r]$ be the current search range. Set $d = \min(2^{2^i}, r - \ell)$. Check if $a$ is in the range $[\ell, \ell + d]$ or $[r - d, r]$ (by comparing $a$ to $a_{\ell+d}$ and $a_{r-d}$). If $a$ is in one of these ranges, say $[\ell, \ell + d]$, do a regular binary search (by choosing the middle point of the range each time) on the range $[\ell, \ell + d]$, until $a$ is found. Otherwise, start the next iteration with the new search range $[\ell + d + 1, r - d - 1]$.

The algorithm continues until $a$ is found.

## 4.2 Analysis for Multiple Predictions

We now state the following theorem regarding the algorithm for multiple predictions. The overhead of using multiple predictions is a $\log m$ factor. The proof is very similar to the proof of Theorem 3.1 and has been deferred to Appendix A.

**Theorem 4.1.** *Given $m$ different distributions, the expected query complexity of the algorithm is* $\log(m) \cdot O(H(p) + \max(\min_{k \in [m]} \log \eta_k, 0))$.

# 5 Experiments

We now present an empirical evaluation of the proposed algorithms on both synthetic and real datasets. Our goal is to show how predictions can be used to improve the running time of traditional binary search approaches. Since our theoretical results are about query complexity, and to keep the results implementation-independent, our main metric will be the number of comparisons performed by each method. Our implementation can be found at https://github.com/AidinNiaparast/Learned-BST.

We compare the performance of the following algorithms:

- **Classic** - The prediction agnostic approach that recursively queries the midpoint of the array.

- **Bisection** - The bisection algorithm recursively queries the median of the predicted distribution (when the predicted probability in the search range is 0, this algorithm queries the midpoint of the array). This strategy is nearly optimal when the predicted distribution is correct [Mehlhorn, 1975]; however, it is not robust to errors in the predicted distribution.

- **Learned BST** - The algorithm described in Section 3. We make one modification, setting the parameter $d$ larger to broaden the search space in the very early iterations, setting d to $\min(2^{8 \cdot 2^i}, r - \ell)$.

- **Convex Combination** - This is a heuristic approach to make the Bisection algorithm more robust. Given a prediction $\hat{p}$ we generate a new distribution, $q = \lambda \hat{p} + (1 - \lambda)u$, where $u$ is the uniform distribution on $[n]$. We then run the Bisection algorithm on $q$. In our experiments, $\lambda = 0.5$ is used.

## 5.1 Synthetic Data Experiments

We begin with experiments on synthetic data where we can vary the prediction error in a controlled environment to show the algorithms sensitivity and robustness to mispredictions.

In this setting, let the keyspace be the integers in $[-10^5, 10^5]$. We then generate $t = 10^4$ independent points from a normal distribution with mean 0 and standard deviation 10, rounding down each to the nearest integer. This results in a concentrated distribution in a very large key space. The $t$ points form the predicted distribution, $\hat{p}$. To generate the test distribution, we proceed in the same manner, but shift the mean of the normal distribution away from 0 by some value $s > 0$. Note that for $s = 100$ the train and test distributions have 0 overlap with high probability. For each value of $s$, we repeat the experiment 5 times and report the average and standard deviation of the costs.

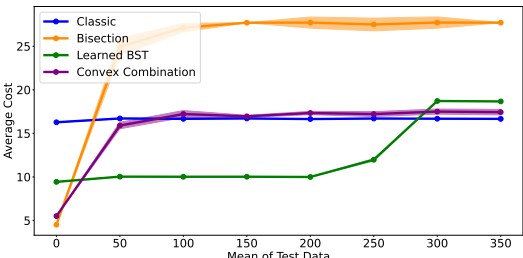

Figure 1: Results for synthetic data experiments. The y-axis measures the average cost (query complexity) of each algorithm and the x-axis measures the amount of shift in the test distribution. The training and test data are regenerated 5 times. The solid lines are the mean and the clouds around them are the standard deviation of these experiments.

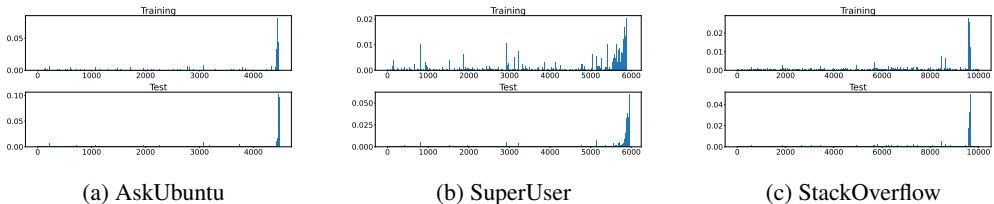

(a) AskUbuntu          (b) SuperUser          (c) StackOverflow

Figure 2: The train and test distributions when $t = 50$ for the three datasets.

Our results for this setting can be found in Fig. 1, where we plot the average search cost (query complexity) of each algorithm against the shift amount for the test distribution. At one extreme, where there is no shift in the test distribution, we observe that all three algorithms which utilize the predicted distribution perform well. Since the bisection algorithm is optimal when the error is 0, it performs the best, as expected, while the Learned BST approach exhibits some overhead due to hedging against possible errors. However, a perturbation to the predicted distribution causes the bisection algorithm to perform worse than classical binary search. Both the convex combination and learned BST algorithms demonstrate a smoother degradation in performance, with our proposed method (learned BST) giving more robust performance to even large shifts in the test distribution. When the erorr becomes very high, then the additional overhead of the learned BST algorithm makes it slightly worse than the Classic baseline.

## 5.2 Real Data Experiments

**Dataset Description.**    In order to test our approach on real-world data, we use temporal networks from Stanford Large Network Dataset Collection[5]. These datasets represent the interactions on stack exchange websites StackOverflow, AskUbuntu, and SuperUser [Paranjape et al., 2017]. In all cases, we use the answers-to-questions dataset, which contains entries of the form $(u, v, t)$, which represents user $u$ answering user $v$'s question at time $t$. In this interaction, $u$ is the source and $v$ is the target user. Our data sequences consist of the source users from each interaction sorted in increasing order of timestamp, and we restrict the dataset to the first one million entries.

**Keys, Predictions, and Test Data.**    For each data sequence, the set of elements in the first 10% of the sequence is used as the set of keys of the binary search trees. Let $A$ be the remaining 90% of the sequence and let $a_1 < a_2 < \ldots < a_n$ be the set of keys. For each element $x \in A$, if $a_i \leq x < a_{i+1}$, we replace $x$ by $a_i$. For $t = 5, 10, \ldots, 50$, we use the first $t$ percent of $A$ as training data and the rest as test data. The training and test data are used to obtain the predictions ($\hat{p}$) and actual access distribution ($p$), respectively. To obtain these distributions we use the normalized frequencies of each key in the training and test data.

For completeness, we show the distributions of the keys when $t = 50$ both for the training set and the test set in Figure 2.

---

[5]https://snap.stanford.edu/data/index.html

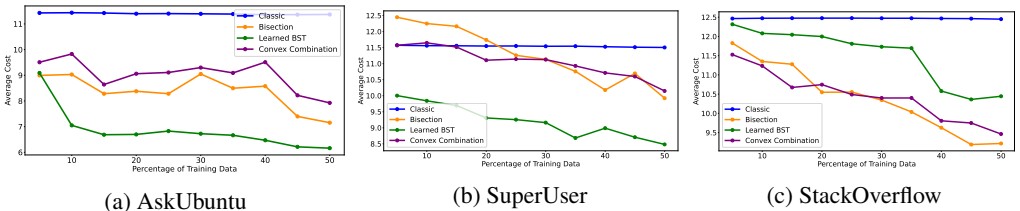

(a) AskUbuntu        (b) SuperUser        (c) StackOverflow

Figure 3: Results for real data experiments. The y-axis measures the average cost of each algorithm and the x-axis indicates the fraction of the dataset used for training

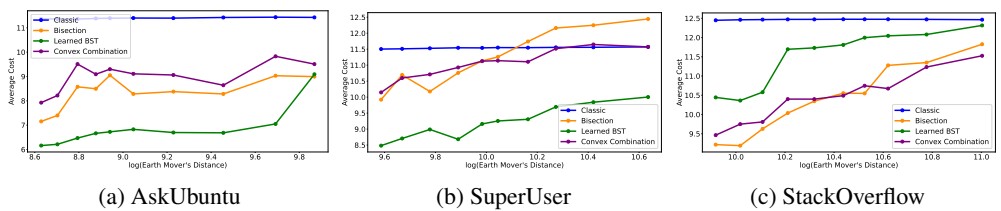

(a) AskUbuntu        (b) SuperUser        (c) StackOverflow

Figure 4: Results for real data experiments. The y-axis measures the average cost of each algorithm and the x-axis indicates the logarithm of the earth mover's distance between $\hat{p}$ and $p$.

We present the results on these experiments in Figures 3 and 4. In Figure 3 we plot the average cost of the algorithms against the size of the training data. As we expect, as the size of the training data increases, the performance of all distribution-dependent algorithms get better, as the distribution error decreases. We make this more precise in Figure 4 where we plot the average cost against log of the EMD error.

We note a few observations. The learning agnostic, Classic, is suboptimal in all but a handful of cases, showing that there is value in using the distribution of the data to improve performance. Second, we validate the theory, showing that the learned BST's performance degrades smoothly as $\log \eta$ increases. Third, the convex combination heuristic is not very effective on real world data, giving only marginal improvements over the bisection method.

Finally, on both AskUbuntu and SuperUser datasets, the learned BST approach performs significantly better than all of the baselines, saving 20-25% comparisons on average. Unlike the Bisection algorithm it is also never worse than the Classic baseline. On the StackOverflow dataset our approach is about 10% worse than bisection method, owing to the distribution being less concentrated around the median. In these cases, the overhead of learned BST is apparent, given that the second phase is unlikely to be fruitful in the first few iterations.

Overall, these results show that the Learned BST method is robust against errors, and performs well against other approaches. Further improving the constant factors so that the learned approach has strong worst-case guarantees and performs well against other learned approaches remains a challenging open problem.

# 6   Conclusion

There has been a growing line of work showing how to improve optimization algorithms using machine learned predictions. Predominately, prior work has leveraged non-probabilistic predictions, despite the fact that most ML systems, such as neural networks, output a distribution.

This work introduces a model where the prediction is a distribution. We show that algorithms can perform better by taking full advantage of the distributional nature of the prediction, and that reduction to a point prediction is insufficient to provide competitive algorithms.

Given the breadth of work in the Algorithms with Predictions area Mitzenmacher and Vassilvitskii [2021], there is a wide variety of open questions concerning how to adapt algorithms to the setting of distributional predictions.

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

# A  Omitted Proofs

*Proof of Theorem 4.1.* With probability $p_i$, the key $a$ that we are looking for is $a_i$. The goal is to bound the expected cost of the algorithm, which is $\sum_{i=1}^n p_i C(a_i)$, where $C(a_i)$ is the cost of the algorithm when $a = a_i$. In each step of the Bisection phase, a binary search is done on the medians of the predicted probability distributions on the current search range. When the binary search is done, for each prediction $k$, the median $t_k$ of the prediction falls outside of the new search range. This means that the probability mass of $\hat{p}_k$ on the new search range has dropped by at least a factor of 2 compared to the initial search range before the binary search. Let $k^* = \arg\min_{k \in [m]} \eta_k$. From the above discussion, the probability mass of $\hat{p}_{k^*}$ in the search range drops by a factor of at least 2 in each step of the Bisection phase, which results in a total drop of at least $2^{2^j}$ in iteration $j$ of the algorithm. The cost of each Bisection step is $\log m$, which makes the total cost of the Bisection phase in iteration $j$ equal to $(\log m) \cdot 2^j$. Let $T_i$ be the first iteration at which $a$ is found, assuming $a = a_i$. The total cost of the first phase of the iterations 0 to $T_i$ is $(\log m) \sum_{j=0}^{T_i} 2^j < (\log m) \cdot 2^{T_i+1}$. Also, the cost of the second phase in each iteration before $T_i$ is 2, and in iteration $T_i$ is at most $2^{T_i}$. So the total cost of the algorithm for iterations 0 to $T_i$ is at most $(\log m) \cdot 2^{T_i+1} + 2^{T_i} + 2T_i = O((\log m) \cdot 2^{T_i})$. We partition the keys based on $T_i$ into two sets, and bound the cost of each set separately. Let $I_1 := \{i : T_i \leq \log(\log(4/p_i))\}$ and $I_2 := \{i : T_i > \log(\log(4/p_i))\}$.

First, we bound the cost of indices in $I_1$ by $(\log m) \cdot O(H(p))$:

$$\sum_{i \in I_1} p_i C(a_i) = O\left(\sum_{i \in I_1} p_i \left((\log m) \cdot 2^{T_i}\right)\right) = (\log m) \cdot O(\sum_{i \in I_1} p_i \log(4/p_i)) = (\log m) \cdot O(H(p)).$$

Now we bound the cost of the indices in $I_2$ by $(\log m) \cdot O(\max(\log(\eta_{k^*}), 1))$. Let $i \in I_2$. We know that during iteration $j$ of searching for $a_i$, in the Bisection phase, the predicted probability mass $\hat{p}_{k^*}$ in the search range decreases by a factor of at least $2^{2^j}$. Therefore the predicted probability mass $\hat{p}_{k^*}$ in the search range $[\ell, r]$ at the end of the first phase in iteration $T_i - 1$ is at most

$$\frac{1}{\prod_{j=0}^{T_i-1} 2^{2^j}} = \frac{1}{2^{2^{T_i}-1}} = \frac{2}{2^{2^{T_i}}} \leq \frac{2}{4/p_i} = \frac{p_i}{2},$$

where the inequality holds because $i \in I_2$. So $\sum_{j=\ell}^r \hat{p}_{j,k^*} \leq p_i/2$. Let $D_i := \min(i - \ell, r - i)$. In the transportation problem corresponding to the earth mover's distance between $p$ and $\hat{p}_{k^*}$, a probability mass of at least $p_i/2$ needs to be moved from point $i$ to the outside of the interval $[\ell, r]$. The cost of this movement in the objective function of the transportation problem is at least $D_i \cdot p_i/2$. Therefore we have $\eta_{k^*} \geq \sum_{i \in I_2} D_i \cdot p_i/2$. In the Binary Search at the Endpoints phase of iteration $j$, we probe indices distance of $d = 2^{2^j}$ around the two endpoints of the search range. Since $a_i$ is not found before iteration $T_i$, we conclude that $2^{2^{T_i-1}} < D_i$, which means that $2^{T_i} \leq 2\log(D_i)$. Let $p(I_2) := \sum_{i \in I_2} p_i$. We have

$$\sum_{i \in I_2} p_i C(a_i) \leq (\log m) \cdot O\left(\sum_{i \in I_2} p_i 2^{T_i}\right) \tag{1}$$

$$\leq (\log m) \cdot O\left(\sum_{i \in I_2} p_i \log(D_i)\right) \tag{2}$$

$$\leq (\log m) \cdot O\left(\sum_{i \in I_2} p_i \log(D_i) + (1 - p(I_2)) \log(1)\right) \tag{3}$$

$$\leq (\log m) \cdot O\left(\log\left(\sum_{i \in I_2} p_i D_i + (1 - p(I_2))\right)\right) \tag{4}$$

$$\leq (\log m) \cdot O\left(\max(\log(\eta_{k^*}), 1)\right), \tag{5}$$

where inequality (4) results from concavity of $\log(\cdot)$ function and Jensen's inequality, and inequality (5) is because of the following

- If $\sum_{i \in I_2} p_i D_i \leq 1$ then we have

$$\log \left( \sum_{i \in I_2} p_i D_i + (1 - p(I_2)) \right) \leq \log(2) = 1$$

- If $\sum_{i \in I_2} p_i D_i > 1$ then we have

$$\log \left( \sum_{i \in I_2} p_i D_i + (1 - p(I_2)) \right) \leq \log \left( \sum_{i \in I_2} p_i D_i \right) + 1 = O(\log(\eta_{k^*})).$$

$\square$

## B  Experimental Setup

We use Python 3.10 for conducting our experiments on a system equipped with an 11th Generation Intel Core i7 CPU running at 2.80GHz, 32GB of RAM, a 128GB NVMe KIOXIA disk drive, and a 64-bit Windows 10 Enterprise operating system. It's worth noting that the cost of the algorithms, i.e., the expected query complexity, is hardware-independent.

