# OpenReview forum: "Binary Search with Distributional Predictions"
_NeurIPS.cc/2024/Conference — NeurIPS 2024 poster_

### Official Review · Reviewer_ud7J · 2024-06-26

**Soundness:** 3
**Presentation:** 3
**Contribution:** 4
**Rating:** 7
**Confidence:** 4

**Summary:**

The main question guiding the paradigm of algorithms with predictions is the following: if we want to solve a given instance of a classical computer science problem, but are given predictions of some form (typically by some machine learning algorithm which has seen a lot of similar problem instances as training data), can we use the predictions to our benefit? By "to our benefit", we want two properties: 1) consistency i.e., if the prediction is indeed very accurate, then we want to fully utilize this, and not suffer the worst-case performance that we would have suffered had we not used the predictions at all and used the default prediction-agnostic algorithm. 2) robustness i.e., if the prediction is fully bogus, then we still want the performance of our algorithm to be no worse than the worst-case performance of the prediction-agnostic algorithm. The classic example in this space is that of performing binary search on a sorted array. Given a key query, we want to find if the key is present in the array. Suppose we have access to a prediction that tells us where we should look at in the array. The classic prediction-agnostic binary search algorithm has a worst case running time of $O(\log n)$. But if the prediction is accurate, note that we can finish in constant time---so we want to utilize this somehow. It turns out that by seeding our binary search at the predicted value, we can achieve this best-of-both-worlds guarantee.

While most of the existing literature on algorithms with predictions uses predictions that are point-valued predictions, this work considers the problem setting where the prediction is a distribution. The justification is that, modern day generative models naturally output whole distributions. The problem under consideration is still that of searching a key in a sorted array of size $n$. But now, we assume that the queries come from a distribution $p$. A classical "median bisection" algorithm due to Knuth (1971) and Mehlhorn (1975) shows that the best expected running time achievable is $\Theta(H(p))$, where $H(p)$ is the entropy of $p$. Suppose the distribution $p$ is heavily concentrated on a few middle elements in the array. Then the classical binary search algorithm takes (expected) running time $\Omega(\log n)$, which is much worse than $H(p)=O(1)$. What if we are given point predictions? The authors give a simple argument that even then, an algorithm that uses such point predictions along with binary search is doomed to suffer a running time of $\Omega(\log n)$.

But what if we are given a predicted distribution $\hat{p}$, which is supposed to be an estimate of $p$? Note that if $\hat{p}=p$, we should in theory be able to obtain a running time of $H(p)$. But we don't know how well $\hat{p}$ approximates $p$. The authors show that it is possible to make use of the predicted distribution $\hat{p}$ to our benefit. Concretely, they give an algorithm that makes use of the explicit form of $\hat{p}$, and has an expected running time of $O(H(p)+\log \eta)$, where $\eta$ is the Earthmover distance between $\hat{p}$ and $p$. Note that when $\hat{p}\approx p$, this achieves a running time of $O(H(p))$ which is what we wanted (and which is the best possible). Also, when $\eta=n$, which is the largest it can be, the guarantee is still $O(H(p)+\log n) = O(\log n)$, since $H(p) \le \log n$. This is in fact the worst-case running time of the classic prediction-agnostic binary search algorithm. The algorithm that the authors propose is a careful combination of classic binary search and the median bisection algorithm (which uses $\hat{p}$ to compute its medians). The analysis of the algorithm is insightful and easy-to-follow.

Is $O(H(p)+\log \eta)$ the best attainable guarantee? Note that even if $\hat{p}=p$, we can't do any better than $H(p)$ by the argument of Mehlhorn (1975). Thus, the question is if the $\log \eta$ factor is necessary. Indeed, the authors show it is, and hence their algorithm is optimal. The authors also propose an extended version of their algorithm which can use multiple predicted distributions, and gets a guarantee with respect to the predicted distribution that has the least Earthmover distance. Finally, the authors demonstrate the efficacy of their algorithm against baselines={classic binary search, classic median bisection} in synthetic simulations as well as real-world datasets. The experimental results are compelling and validate the theory to a large extent, showcasing the benefit of using the full-power of the distributional form of the prediction at hand.

**Strengths:**

As mentioned, this paper departs from the usual algorithms with predictions paradigm which assumes access to point-valued predictions, but instead considers the setting where the prediction is a distribution in itself. The authors make a strong case for the richness of this seemingly more general problem setting: even for the textbook example of searching in a sorted array, the classic binary search algorithm with a point-valued prediction no longer attains the best achievable performance, and a non-trivial algorithm, that makes explicit use of the full form of the distribution-valued prediction, is necessary to achieve performance gains. Moreover, it is commendable that the algorithm presented by the authors is provably optimal. The work of the authors makes a strong case for studying other classical algorithms-with-predictions problems from the literature under the setting of distribution-valued predictions. The theory is also well-supported by the experiments.

**Weaknesses:**

The main concern I have is with the Lower Bound in section 3.2. See Questions below. I would also tone down the claim of "*initiating* the study of algorithms with distributions as predictions" (line 6, line 59), especially since, as the authors themselves mention, such a setting was also studied by Angelopoulos et al. (2024).

---

Minor/typos: \
Line 60: It seems that there is a slight change in problem setting here for the algorithm of Knuth (1971), where now we think of queries being drawn from a distribution $p$. This should be stated beforehand, otherwise saying things like "median of the distribution" does not make sense, when no distribution has been introduced. Also, $p$ is not introduced notationally. \
Line 184: $S$ is not introduced notationally. I believe it should be the probability mass on $[l,r]$ i.e., $S=\sum_{i=l}^r \hat{p}_i$ (correct me if I am wrong) \
At a lot of places, parenthetical citations \citep{} ought to be used (e.g., lines 117-118 in the related work, etc.)

**Questions:**

I am a little confused by the proof of the lower bound. Where exactly are you using the form of the predictions $\hat{p}$ in your argument? In line 245, you simply say that, "the best thing that an algorithm could do is build a balanced binary search tree on $[\eta]$." It seems to me that this sentence stems from the way the distribution over $p$'s is constructed, and is agnostic to the specific form of $\hat{p}$. Put another way, could you point out what is flawed in the following proof, which gets a $\Omega(\log{n})$ lower bound?

Take $\hat{p}$ to be uniform on all of $[n]$. The distribution over true instances $p$ (which are distributions over keys) is simply: draw $i^* \in [n]$ uniformly at random, and set $p_i=1[i=i^*]$. Then, by the reasoning in line 244-245, any deterministic algorithm will have at least $\Omega(n)$ keys such that the cost on these keys is $\Omega(\log n)$, because again, "the best thing an algorithm could do is build a balanced binary search tree on $[n]$". Hence expected cost is $(1/n)(\Omega(n))(\Omega(\log n)) = \Omega(\log n)$. Proof concluded by noting EMD between $\hat{p}$ and $p$ for any $i^*$ is at most $n$.

Depending on clarification about this, I will either increase/decrease my score...

**Limitations:**

I do not foresee any limitations that the authors have not addressed.

---

> ### Author Rebuttal · Authors · 2024-08-06
>
> We thank you for the points you have brought up in your review.  We will incorporate the minor comments you raised into the paper and clarify the notation.  As to your main comment about the lower bound, we will improve the rigor and also hope that the following discussion clarifies the lower bound for you.
>
> It is true that the construction you gave gives a lower bound of $\Omega(\log n)$, when the earth mover distance between $\hat p$ and $p$ is $\approx n$.  But one might worry that this lower bound is only possible because of the large EMD; a priori, it could be the case that $\log \eta$ is a lower bound when the EMD is extremely large, but once it is smaller (we have a more accurate prediction) the dependence can be improved.  Since we are most concerned with the case of reasonably accurate predictions, if this were the case then our lower bound would not be particularly meaningful.      To handle this, the lower bound we present in the paper generalizes this idea to handle any value of $\eta$, instead of just $\eta = n$.
>
> In other words, the theorem we prove is the following: “For any $\eta \in [0,n]$, any comparison-based algorithm must make $\Omega(\log \eta)$ queries on some instance where $H(p) = 0$ and the Earth Mover’s Distance between $p$ and $\hat p$ is $O(\eta)$.”
>
> This is a stronger lower bound than the one suggested by the reviewer, since it holds for all $\eta$ rather than just for when $\eta = \Theta(n)$.  Moreover, since the bad instances have $H(p) = 0$, it shows that there must fundamentally be a $\log \eta$ dependence on the Earth Mover’s Distance; this does not only happen when it can be absorbed by the dependence on the entropy.  We will calrify these points upon revision.
>
> Regarding Angelopoulos et al. (2024), this contemporary and independent paper studies how to obtain an optimal tradeoff between consistency and robustness with distributional predictions. However, their solution space explored is considerably more limited than ours, consisting of geometric sequences with a multiplicative ratio of 2, each characterized by its starting point. Additionally, the gap between consistency and robustness is relatively small (4 vs. 2.77), implying less room to leverage predictions. Finally, their bound analysis is restricted to cases where the error is sufficiently small.
> In contrast, our work demonstrates how a binary search algorithm can compare generally to Earth Mover's Distance (EMD) and a lower bound on the optimum for any predicted distribution. As a result, we develop novel algorithmic solutions that build upon a close connection to EMD. We will ensure that this previous work receives proper credit.

---

> > ### Comment · Reviewer_ud7J · 2024-08-07
> > **Response to rebuttal**
> >
> > Thanks for your response and clarification. It would be great if you could update the lower bound proof and statement as you mention above. Please also include the discussion about Angelopoulos et al. (2024) in the revision. I have updated my score from 6 -> 7. I maintain that this is a strong contribution, and deserves to be accepted. Great job!

---

### Official Review · Reviewer_EXym · 2024-07-12

**Soundness:** 2
**Presentation:** 4
**Contribution:** 3
**Rating:** 6
**Confidence:** 4

**Summary:**

The paper studies the problem of binary search for an item in a sorted array in a learning-augmented setting, where the element to search for is drawn from some unknown distribution, and the algorithm has access to a prediction of this distribution. This contrasts with the majority of work in the learning-augmented setting (aka algorithms with predictions), most of which assume that the prediction is an exact value rather than a probability distribution.

The main result is that there is an algorithm performing this task in $O(H(p)+\log \eta)$ expected time, where $H(p)$ is the entropy of the true distribution and $\eta$ is the Earthmover distance between the predicted and the true distribution. This extends the known $O(H(p))$ result from the case where the exact distribution is known ($\eta=0)$ to the case where only a prediction of the distribution is known. A matching lower bound on the dependence on $\eta$ is also given.

The basic skeleton of the algorithm is to run the known algorithm for the case of known distribution. As the search interval shrinks during this process, it is repeatedly (in exponentially growing steps) checked whether the search item is close to the left or right end of the current search interval. If so, the algorithm concludes with a classical binary search at the respective (left or right) end.

The algorithm is further extended to the setting where $m$ predictions are given, losing a factor of $O(\log m)$ in the guarantee with respect to the best of the given predictions. The paper concludes with an experimental evaluation, yielding results in line with the theory.

**Strengths:**

- Extending learning-augmented algorithms to the more realistic case where the prediction is a distribution, which is lacking in results at the moment
- Tight bounds for a fundamental problem
- Clearly written

**Weaknesses:**

- It is not clear whether the bounds hold for the running time, as claimed, or only for the number of comparisons. The main algorithm has a step where the index of a median element is identified, which the analysis does not seem to be account for. Moreover, the portfolio algorithm has a step where m indices are sorted, which would seem to take time $O(m \log m)$ and increase the bound by an additional factor $m$. One might treat this as preprocessing time that is only incurred once before many binary searches are conducted. However, for such preprocessing to become negligible would seem to require many subsequent binary searches, yielding many samples from the true distribution, which would seem to allow approximating the true distribution very precisely rather than relying on a (collection of) prediction(s).
- The experimental evaluation is restricted to settings where the distributions are highly concentrated around a single value.

Considering the authors' comments in the rebuttal to these two points, I'm increasing my score from 5 to 6.

**Questions:**

1. Please clarify whether/how the claimed running time bounds can be attained, or if the bounds hold for the number of comparisons instead.
2. I suggest expanding the discussion of related work. For example, the paper "Beyond IID: data-driven decision-making in
heterogeneous environments" by Besbes et al seems to consider a similar setting where an algorithm has access to a prediction of a distribution.
3. The text says that Figure 2 is for t=5, but the Figure caption says t=50 instead.

**Limitations:**

Not specifically discussed. A discussion about the possible weaknesses mentioned above would be useful.

---

> ### Author Rebuttal · Authors · 2024-08-06
>
> Thank you for your thorough review.  We hope we can address each of your comments below.
>
> 1. Running time versus comparisons.
>
> We are sorry for the confusion; putting “running time” in our theorem statements was a mistake.  We meant query complexity (or number of comparisons), as the reviewer realized. See the discussion in response to reviewer kf39.
>
> 2. The experimental evaluation is restricted to settings where the distributions are highly concentrated around a single value.
>
> The distribution needs to be reasonably concentrated for our algorithm to outperform binary search (the best worst case algorithm). If the distribution is too close to uniform, standard binary search is the better thing to do (although, as the theory shows, even if the distribution is close to uniform we will not be too far from optimal).
>
> We remark that our algorithm outperforms both the classic binary search and the bisection algorithm on several real data sets as shown in the paper. Further, the distribution does not need to be concentrated in order for our methods to see gains. For instance, in experiments below, we show improvements on bimodal distributions.
>
> 3. I suggest expanding the discussion of related work. For example, the paper "Beyond IID: data-driven decision-making in heterogeneous environments" by Besbes et al seems to consider a similar setting where an algorithm has access to a prediction of a distribution.
>
> Thank you for suggesting the work by Besbes et al. A key point of difference is that our algorithm does not need to know the error of the prediction ahead of time. In more detail, the work of Besbes et al. considers stochastic optimization problems such as the newsvendor, pricing, and ski-rental problems under distribution shift and analyzes the asymptotic performance of the sample-average-approximation (SAA) policy in this setting.  For pricing and ski-rental, they show that SAA is not robust to small distribution shifts, but that shifting by a small amount (depending on the amount of distribution shift) in the appropriate direction is robust.  In contrast, we make no assumption about where the distribution prediction comes from and our algorithm does not need to know how incorrect the distribution is to remain robust.
>
> 4. The text says that Figure 2 is for t=5, but the Figure caption says t=50 instead.
>
> Thank you for spotting the typo, the correct training data percentage is $t=50$. We will fix the discrepancy.
>
> Additional experiments:
>
> We have included in this rebuttal new synthetic experiments on bimodal distributions. The setting is similar to that of section 5.1, with the difference that both the predictions and the actual access distributions are bimodal. This is the details of the setting:
> The keyspace is the integers in $[-10^5,10^5]$. To generate the predictions, we generate $10^4$ independent points from each of the following two distributions (rounding down each to the nearest integer):
>
> a. A normal distribution with mean 0 and standard deviation 10.
>
> b. A normal distribution with mean $d$ and standard deviation 10.
>
> We ran the experiments for different values of $d$, which is the distance between the means of the two normal distributions (or equivalently the distance between the two peaks of the final bimodal distribution). For each value of $d$, to generate the actual access distribution (i.e., the test data) we follow the same procedure, but we shift the two peaks of the distribution by some value $s>0$, i.e., the test data is generated by sampling $10^4$ points independently from each of the following two distributions:
>
> a. A normal distribution with mean $s$ and standard deviation 10.
>
> b. A normal distribution with mean $s+d$ and standard deviation 10.
>
> For each value of $d$ and $s$, we repeated the experiment 5 times and calculated the average number of comparisons of each algorithm. Here we report the results for $d=100$.
>
>
>
> | | s=0 | s=50 | s=100 | s=150 | s=200 | s=250 | s=300 | s=350 |
> | --- | --- | --- | --- | --- | --- | --- | --- | --- |
> | Classic | 16.47 | 16.71 | 16.65 | 16.71 | 16.65 | 16.69 | 16.66 | 16.69 |
> | Bisection | 5.50 | 20.96 | 17.01 | 27.39 | 28.13 | 28.89 | 28.52 | 28.70 |
> | Learned BST | 9.51 | 9.86 | 9.99 | 10.01 | 12.43 | 14.37 | 15.51 | 18.69 |
> | Convex Combination | 6.50 | 16.09 | 12.07 | 17.30 | 17.62 | 17.69 | 17.67 | 17.69 |
>
> As the results show, our algorithm outperforms the baselines even when there is a large shift in the test distribution. Similar results were obtained for different values of $d$, including $d=50,200,1000$.

---

> > ### Comment · Reviewer_EXym · 2024-08-12
> >
> > Thank you for your response. I'm increasing my evaluation from 5 to 6.

---

### Official Review · Reviewer_mBjQ · 2024-07-12

**Soundness:** 3
**Presentation:** 4
**Contribution:** 3
**Rating:** 7
**Confidence:** 4

**Summary:**

In this paper, the authors study the classical problem of binary searching over a sorted array in the learning-augmented setting, with distributional advice. Binary search in both the learning-augmented setting with point advice and the classical setting with known query distribution is well studied, but the authors are the first to study binary search in the learning-augmented setting with distributional advice. The authors argue that distributional advice is more natural than point advice as outputs of machine learning models such as neural networks, and show that distributional advice can actually perform asymptotically better than point advice.

Specifically, the authors consider the model in which the target element of the binary search is drawn from an unknown distribution, and aims to minimize the expected cost of searching the element. The search algorithm is also given a prediction on the underlying distribution; the prediction error is measured in the earth mover distance between the prediction and the true distribution. Their search algorithm is not trivial by any means, but very intuitive: On a high level the algorithm executes in iterations, where in each iteration it conducts median search based on the predicted distribution a set number of times depending on the current iteration, and checks both ends of the search range, finishing with a normal binary search if the target is near the endpoints. If the target is not found near the endpoints, start the next iteration on the middle range.

Their analysis is simple and shows that in expectation the algorithm uses $O(H(p) + \log \eta)$ comparisons, where $H$ is the binary entropy function, $p$ is the true distribution, and $\eta$ is the earth mover distance between $p$ and the prediction $\hat p$. As corollaries, they show that the algorithm can adapt to multiple predictions, and conduct experiments to show the performance of their algorithm on real and synthetic data sets.

**Strengths:**

The paper is very cleanly written and technically sound, and really does not have a lot to complain about. I was initially somewhat skeptical to see a significant new result about learning-augmented binary search, which is a very well-studied subject, but the authors managed to bring me a welcomed surprise.

Their "claim to fame" is on the novelty of the advice model, using a distributional prediction as opposed to a traditional point advice. While I have some minor questions about their model, they do indeed achieve a very impressive performance, and back it up with a matching lower bound analysis. A large part of projects in the field of learning-augmented algorithms, in my understanding, is about finding the right form of advice to use in conjunction of a traditional algorithm, and using more probabilistic, distributional advice is a direction I am quite interested in as well.

**Weaknesses:**

I have no major complaints with the paper, but in my opinion, the author's usage of distributional advice is, on a high level, similar to how other learning-augmented use their respective form of advice. The author's search algorithms rely on the median of the distribution to conduct bisection of the search range, which is in of itself not a probabilistic operation; It also relies on the fact that the input to the algorithm is an element drawn from a distribution. One can arguably draw parallel to other non-distributional learning-augmented algorithms for online problems that uses predictions on its input sequence, which is also unknown to the algorithm beforehand. There are also scenarios in both online and offline algorithms where the input may not be drawn independently, or may not be drawn from a distribution at all, where non-distributional learning-augmented algorithms may still hold value.

I do not think this invalidates the author's work - I still find the paper really impressive, but personally I do not find the divide as great as the author painted.

**Questions:**

As I've outlined above, I don't find division between point predictions and distributional predictions as large as the authors painted. It is a personal opinion though, so I would like to hear the author's thought on this matter.

It would be interesting to see what other problems can distributional advice excel on. Binary search is still a relatively simple problem, so application of the same methodologies on problems with more/less structure would be much more convincing.

**Limitations:**

The authors discuss their limitations properly. There are no ethical concerns.

---

> ### Author Rebuttal · Authors · 2024-08-06
>
> We agree that it is natural to adapt the algorithms with predictions model to a distributional prediction by simply using the median as the predicted value.  However, if the benchmark is the optimal solution on the true distribution, using the median as a point prediction will not give strong guarantees compared to optimal. We prove an even stronger statement in Section 2.1 –  a separation between the point prediction setting and the distributional prediction setting.  In other words, in Section 2.1 we show that if you restrict yourself to a point prediction (whether it is the median of the given distributional prediction or something else), you simply must pay more than if you actually use the information given by the distributional prediction.  So while in some settings there might not be much difference between point and distributional predictions, in our setting (the most basic prediction setting!) there is.
>
>
> This paper takes a step in understanding a rich and largely unexplored area. That is, how can one design and analyze an algorithm for a problem when (1) the true distribution is stochastic and the benchmark is the optimal solution and (2) the algorithm is given a noisy distribution.

---

> > ### Comment · Reviewer_mBjQ · 2024-08-08
> >
> > Thank you for your response. I understand why point predictions do not work as well for binary search compared to distributional predictions, and that is not the point of my comment in the weakness section. I am more pointing out that the way this distributional prediction is used in your learning-augmented algorithm does not differ as much in my opinion as the introduction portrays it to be: using the (deterministic) information extracted from a distributional prediction to make deterministic actions (bisection on the search range). This is largely personal opinion so I would like to hear your thoughts on this.
> >
> > Your response does raise an interesting questions also: what makes a distributional prediction champion over point median predictions? What additional properties of the predicted distribution helps provide this advantage? I think this will be a very interesting point to discuss, if it hasn't been already.

---

> > > ### Author Response · Authors · 2024-08-08
> > > **Further clarifications**
> > >
> > > Thanks for the clarification.  We agree – on one hand this is similar to how most algorithms with predictions literature uses predictions. That is, the algorithm changes its control flow deterministically based on some information extracted from the prediction. On the other hand, note that the distribution that the algorithm considers is updated from phase to phase, since we need to look at the median of the distribution conditioned on the endpoints of the current search space. So in some ways this is like the online setting mentioned by the reviewer, in that we use a “new” prediction at each step.  But this is not an online problem, and our “new” prediction is just a computational transformation of the old prediction.  So we completely agree that there are many similarities!  But there are also important differences that only arise when considering distributional predictions.
> > >
> > > We also agree that trying to formally understand what makes a distributional prediction better is an interesting open question. For some intuition in the binary search problem, consider the setting where the truth is multimodal (e.g. the distribution has mass of 1/(k+1) at points 0, n/k, 2n/k, …, n}, for intermediate values of k). Here a point prediction will necessarily drop a lot of information, whereas a distribution prediction will preserve the richness of the input space.  This is the essence of our lower bound from Section 2.1, and we believe that this points towards what makes distributional predictions more powerful.
> > >
> > > Overall, we don’t believe that distributional predictions are a silver bullet and should be used in all situations – we (again) completely agree that with the reviewer that “non-distributional learning-augmented algorithms may still hold value”, and, in fact, there are likely many settings where the non-distributional view is at least as useful as the distributional.  But the simple binary search example seems to imply that in the right context, distributional predictions bring a lot of additional power.  Understanding exactly which problems and contexts are a good fit for distributional predictions is a fascinating set of open questions.

---

### Official Review · Reviewer_kf39 · 2024-07-14

**Soundness:** 3
**Presentation:** 1
**Contribution:** 3
**Rating:** 5
**Confidence:** 5

**Summary:**

This paper proposes a learning-augmented algorithm for searching in a sorted array. Different from all previous learning-augmented algorithms, it takes in distributional predictions. The main result is an algorithm with query complexity $O(H(p) + \log \eta)$, where $H(p)$ is the entropy of the true distribution and $\eta$ is the Earth Mover's distance between the true and predicted distributions. The paper also includes proofs to show the theoretical optimality and experiments to validate the practical usefulness.

**Strengths:**

- The paper follows the recent line of work on "learning-augmented algorithm", or "algorithm with predictions". This is a promising new direction that tries to combine the theoretical soundness of classic algorithms with the learning ability of machine learning algorithms.
- The section on theoretical analysis (though the ideas are simple) is effective.
- The paper includes experimental results to back up the theory. The experimental settings are diverse. The performance of the proposed algorithm is strong compared to all the baselines.

**Weaknesses:**

- I think the presentation of the paper can be greatly improved. To list a few points:
    - in line 50, the sentence "That is, the prediction is a single (potentially high-dimensional) point (or maybe a small number of such points)." is hard to read for me.
    - in line 56, the sentence "Or, can we in fact do better by taking full advantage of the entire predicted distribution." should end with a question mark.
    - the abusive use of the word "essentially" greatly weakens the soundness of the paper (For example, in lines 66, 70, 78 87, 88). The expression "essentially optimal" should be clarified with collaboration on complexity and constants.
- I think several key works in the field of learning-augmented algorithms are missing, which makes it hard to position this paper in the correct context. For example, I think the algorithms proposed in "Learning-Augmented Binary Search Trees" by Lin et al should be at least discussed and even compared against (now, this paper is only mentioned as a very general reference for learning-augmented data structures).
- It is not straightforward to me why the techniques used in the proposed algorithm are novel and not hard to come up with. I encourage the author to make a clearer point on "which components of the proposed algorithm are novel and different from existing techniques".

**Questions:**

- The algorithm in [Mitzenmacher and Vassilvitskii, 2021] that searches in a sorted list with predictions receive separate predictions for each query. Is this also the case for the setting discussed in this paper?
    - If so, as I understand, the proposed algorithm needs to rebuild the binary search tree every time it receives a new query along with its distributional predictions. Then, this would lower bound the time complexity to answer each query with $O(n)$. Is that correct?
- I am confused by the reference to "binary search tree" in the paper (even in the title). Does the proposed algorithm actually require building a binary search tree in its specification and implementation? Why does section 3 not contain any explanation related to the binary search tree?

**Limitations:**

Most of the limitations are discussed in the weakness section and questions section. In summary, I think the presentation of this paper is not clear enough. This creates difficulties for me to: 1. position the proposed algorithm in the context (of other recent learning augmented searching algorithms). 2. evaluate its novelty. 3. even understand its technical details (as mentioned in the questions section). If the author could provide a better explanation of the above-mentioned points, I would consider raising my score.

---

> ### Author Rebuttal · Authors · 2024-08-06
>
> Thank you for the insightful comments.  As the reviewer astutely observed, the bounds we claim are on query complexity, rather than running time. We apologize for conflating the two and we will make this more precise.
>
> Query complexity is a standard metric when studying data structures. The number of queries is often a more informative measure of complexity.  For example, it could be the case that each comparison requires an I/O to disk (or network!), or a comparison may require an experiment that needs to be performed, so the rest of the computation has negligible cost.
>
> As the reviewer points out, there are two potential ways to interpret the results.
> - Model 1: Queries arrive over time and there is a distribution over locations which arrives with each the query.  This is a strict generalization of [MV 21], where the prediction is a single location in the array. In their model, the location is error prone and in ours the distribution over locations is error prone. Our goal is to construct an effective search strategy given the query and the prediction.
> - Model 2:  there is a single (unknown) distribution over queries and the goal is to design a single binary search tree when given a (possibly erroneous) prediction of this distribution.  Rather than generalizing [MV 21], this generalizes classical work on optimal binary search trees ([Mehlhorn 1975]). So in this model, our work can be thought of as computing an optimal BST that is robust to a distribution shift.  Surprisingly, given the classical nature of computing optimal (or near-optimal) BSTs, this simple question of “what if my distribution is incorrect?” has not been considered in the data structures and algorithms literature.
>
> From the perspective of query complexity, these models are identical: one can treat the prediction in model 1 as the given distribution in model 2.  Any search strategy (from model 1) is a binary search tree (from model 2), and vice versa.  We will discuss this equivalence and clarify all of our claims to be about query complexity in the camera-ready. We note that the revision is minor (including the above discussion and slight changes in theorem statements).
>
> 1. [MV 21] … receives separate predictions for each query. Is this also the case … in this paper?
>
> Yes: we are given the prediction with the query---model 1 described above.  If instead we have a global distribution over queries (model 2), then the prediction comes *before* all of the queries, and constructing the BST is just preprocessing.
>
> 2.  On the need to rebuild the binary search for every query … lower bound the time complexity to answer each query with $O(n)$.
>
> Yes and no.  Yes, the actual time complexity would be larger than the query complexity. This is in some sense unavoidable due to the richness of the predictions.  But no, we do not actually build the binary search tree for each query.  Rather, we give a search algorithm which uses comparisons, where the number of comparisons is bounded by our theorems.  Of course, any search strategy is also a binary search tree.  So we can think of there being an “implicit” binary search tree that our search algorithm is using, even though it is not actually explicitly constructing a binary search tree.  If our focus were model 2, where we want to actually construct a binary search tree, then we would explicitly construct it (but would only do this once as preprocessing, since in that model there is a global distribution rather than one per query).
>
> 3. Comparison to prior work.
>
> The work by Lin et al. is the first foundational work on how to use predictions for data structures, and is related to the second model of our results (discussed above). However, their theoretical results are somewhat narrow in scope: they predict the rank ordering of the elements rather than their full distribution, and their measure of error is specifically tailored to distances on permutations (it only corresponds to a distance in distribution under the assumption that the distribution is Zipfian). Moreover, they make an assumption that the error is small (the values of $\epsilon$ and $\delta$ are constant).  One of the goals of our work was to relate performance to a standard error measure between two distributions (earthmover distance), without any a priori assumptions on its value.
>
> Because their prediction metric is different (orderings vs full distributions), Lin et al. cannot compare their performance to that of the optimal binary search tree, and must settle for comparing to the optimal treap. As such, our results are not directly comparable with theirs.
>
> 4. Our contributions.
>
> A major contribution of the paper is not in the techniques themselves, but in showing that the current literature on learning augmented algorithms that does not consider distributional predictions is suboptimal.  Specifically, as we discuss in the introduction, since modern ML algorithms naturally output a distribution, they are a more natural type of prediction.  Second, as we show in Section 2.1, distributional predictions are **strictly** more powerful than point predictions, so if we want optimal performance we should actually use the full distribution rather than flattening it to a singleton.
>
> While the results and the specific techniques may appear simple in hindsight, It is not a priori clear how to combine the doubling search of [MV 21], which is robust to errors but cannot handle distributions, with the classical distribution-based algorithm achieving H(p) query complexity, which is not robust to errors in the distribution.
>
> Overall, our work studies distributional predictions in a simple setting in order to ease the formalization of the model and emphasize the differences from point predictions. Now that we have demonstrated the importance of distributional predictions, we believe that there will be significant amounts of followup work expanding the algorithms with predictions literature to the case of distributional predictions.

---

> ### Comment · Reviewer_kf39 · 2024-08-11
>
> Thank you to the author for their detailed rebuttal and explanations.
>
> - After changing all occurrences of "time complexity" to "query complexity," I now understand the basic models in the paper. I agree that query complexity is an important metric for search algorithms and data structures.
> - I found the explanations for models 1 and 2 very helpful and believe they should be included in the paper. These explanations provide the necessary context to understand the algorithm, which I previously lacked.
> - I appreciate the clarification on the differences between this paper and the work by Lin et al. While these two works are not directly comparable, I strongly recommend including this discussion in the related work section. The fact that both focus on augmenting binary search trees with learnable advice makes this discussion essential.
> However, I have some remaining concerns about the paper:
> 1. **The discussion of basic baselines seems insufficient.**
> 	On line 173, the author uses the two-point distribution to claim that "converting pˆ to a point prediction and then using the algorithm of Mitzenmacher and Vassilvitskii [2021] as a black box is doomed to failure." However, one immediate idea for this example is to run two MV21 searches (as "black boxes"!) in parallel, starting at the two points, and the key could be found in constant queries. I believe this is a straightforward way to generalize MV21 to distributional settings, and the given example is too simple to demonstrate that this algorithm does not work. The author should include a stronger example to illustrate the suboptimality of "running a few MV21 in parallel."
> 2. **I still find the use of "binary search tree" somewhat confusing.**
> 	While I understand that every search algorithm can be formalized as a BST, I don't see how this phrasing clarifies the algorithm. Just as MV21 can be described as a BST, it is more naturally and simply described as a search algorithm. In the case of the proposed algorithm, I agree that the Bisection phase is similar to descending a balanced BST, but the second phase seems more like checking the left and right boundaries. Therefore, I believe the algorithm and the paper's title should be something like "Searching with Distributional Predictions." Does the author agree?
> 3. There is a lack of justification for the novelty of "algorithms with distributional prediction."
> 	In the author's response, they express the potential of distributional predictions as a major contribution of their paper. However, as the author pointed out in the related work section, the idea of using distributional predictions is not new, and basic tasks like ski-rental have been studied by Diakonikolas et al. [2021]. Therefore, I question the author's claim that "we believe that there will be significant amounts of follow-up work expanding the algorithms with predictions literature to the case of distributional predictions."
> 4. A new question (minor): In line 303, why set the exponential coefficient of $d$ to 8? If this is purely a result of hyperparameter tuning, is there any justification for why this would be preferable in all cases (rather than overfitting to the experiments)?
> 5. Results on time complexity.
> 	I understand that the proposed algorithm has different time and query complexities, but I think the results could be strengthened and made more applicable if the author also addresses the time complexity of the proposed algorithm. This is especially relevant in model 2, where all queries share the same distributional prediction.
>
> Overall, I agree that the algorithm proposed is interesting and, as a result, I have changed my rating from 3 to 5. However, I think the paper requires revision (at the very least, changing all mentions of "running time" to "query complexity" in the main algorithms and major results, adding model 1&2 explanation, and adding related works) to be publishable. I hope the author can further address the concerns I have raised above.

---

> ### Author Response · Authors · 2024-08-12
>
> Thank you again very much for the feedback!  We will certainly include the discussions of model 1 and 2, and more comparison to Lin et al., in the paper as you suggest.
>
> > “The discussion of basic baselines seems insufficient”
>
> We agree that the proof of our specific claim, that one cannot black-box reduce to a single point prediction, does not directly imply that one cannot reduce to *collections* of point predictions.  However, it is relatively straightforward to generalize our proof to this stronger claim – instead of a 2 point distribution, consider a multi-modal distribution with $d = \log^2 n$ points. The method suggested by the reviewer, of doing a black-box reduction to MV for each of a collection of point predictions (by running an instance of MV in parallel for each of them), fails for this example.  Either there is no prediction for one of the $d$ points, in which case every instantiation of MV will do many queries, or we do have a prediction for each of the points, and thus, when we run them in parallel we would probe all $d$ of them, again resulting in suboptimal query complexity. Developing an algorithm that can handle such multi-modal distributions (and generalizations) in an optimal manner is a significant part of our contribution.
>
> > “I still find the use of "binary search tree" somewhat confusing.”
>
> We completely agree that it would be best to change the title of the paper as you suggest.
>
> > “There is a lack of justification for the novelty of "algorithms with distributional prediction."
>
> We will include a more detailed discussion of Diakonikolas et al. We would like to highlight that our use of distributions is different. Specifically, the work of Diakonikolas et al. does not consider what happens when the distribution is erroneous (a major theme of our work), rather their focus is on minimizing the number of samples from the true distribution that they need. Since predictions are often erroneous, there is a dire need to make sure our usage of predictions is robust. This is what we focus on in this work, and believe would be of interest to the general community.  Of course, the empirical distribution of samples can be viewed as an erroneous prediction of the true distribution, but our setting is far more general, and allows for general distributional predictions with general errors (measured by EMD).  So we do believe that our setting is novel and will lead to significant follow-up work.
>
>
> > “A new question (minor): In line 303, why set the exponential coefficient of $d$ to 8? If this is purely a result of hyperparameter tuning, is there any justification for why this would be preferable in all cases (rather than overfitting to the experiments)?”
>
> Setting the exponential coefficient of d to a small constant is preferable for improved empirical results, and does not change the asymptotic complexity. To see why, recall that the algorithm explores segments of length $2^{2^i}$ in the $i$-th iteration. When i is very small, these segments are very small, making the iterations overly fast and unlikely to succeed. We found that setting the length to $2^{8 \cdot 2^i}$ allowed us to balance this trade-off better. The exact setting of the exponential coefficient is not important (4 and 16 worked almost equally well). Also, see the further experimental results in our response to Reviewer EXym which take place on bimodal instances.
>
> > “Results on time complexity.”
>
>  We note that in model 2, the time complexity is basically equivalent to the query complexity.  In this model, there is a single distribution over queries, so the time to build our BST is just preprocessing – we expect to answer far more queries than the time it takes to build the BST (as long as the preprocessing time is at least somewhat reasonable, as ours is).  And then once the BST is built, the time it takes to search in it is essentially equal (up to constants involving following pointers) to the query complexity that we analyze.  So in this setting, the query complexity *is* the time complexity!

---

> ### Comment · Reviewer_kf39 · 2024-08-13
> **Interesting algorithm but significant change in rebuttal**
>
> Thank you to the authors for their response! It has clarified most of the issues. I suggest that the authors add discussions on the $log^2$ multi-modal distribution, the differences compared to Diakonikolas et al., and some commentary on time complexity in the next version of the paper. These points were previously unclear.
>
> **In summary, I believe this paper presents a new algorithm with sufficient theoretical analysis and experiments. If presented clearly, it is worthy of publication at NeurIPS. However, I have a major concern about the presentation: the edited version after the rebuttal might be too different from the originally submitted version.** I found the original version difficult to understand, and only after the authors promised to change or add significant content in the rebuttal discussion, did I find the presentation acceptable. But this level of change during the rebuttal period may be too substantial. For example, the authors agreed with me to change the paper’s title; they also admitted that the original references to "time complexity" were incorrect and will be changed to "query complexity." However, reviewer ud7J, who recommended acceptance, understood the paper completely based on the "time complexity" framework. This discrepancy has led to a misunderstanding in the reviewers' comments. Therefore, I will maintain my rating at 5 and leave it to the area chair to decide whether this level of change during the rebuttal phase is too extensive.

---

### Decision · Program_Chairs · 2024-09-25

**Decision:**

Accept (poster)

**Comment:**

This paper studies the query complexity of searching through a sorted array when the keys are drawn from a given distribution which is approximately known. An algorithm is proposed which interleaves two approaches: the classic binary search algorithm and the "median search" approach from [1,2]. The *query complexity* of the proposed algorithm is shown to be $O(H(p)+\log(\eta))$ where $H(p)$ is the entropy of the ground truth distribution and $\eta$ is the Wasserstein distance between the estimated sampling distribution and the ground truth distribution.

There is a consensus between all four reviewers  (and myself) that this is an interesting and novel problem setting and algorithm which is well deserving of publication. However were some concerns, especially from reviewer kf39, regarding the presentation. I partially agree with the reviewer: on the one hand, I did find the introduction and abstract difficult to parse without jumping straight to the more mathematical "preliminaries" section, and on the other hand, it is also the case that the authors have **incorrectly used** terms such as "**time complexity**" instead of "query complexity" throughout the paper. They have admitted their mistake during the rebuttal and discussion with reviewers ud7J and kf39 and promised to modify the paper accordingly. Reviewer kf39 argues that the required changes might be too much and that the paper might need a fresh round of reviewing. I think this is a subjective question, but my professional opinion is on the side of the authors: removing any explicit mention of time complexity in the theoretical results can be achieved before the camera ready deadline. I also would not find it particularly shocking if some of the *experimental* results demonstrated superior performance in terms of runtime, even if the theory itself only applies to the query complexity, as long as the distinction is made clear.

 There is also enough time for the authors to improve the presentation of the work, and I strongly encourage the authors to do so: this paper's contribution was clearly appreciated by many reviewers and has the potential to be of great interest to the community, if written in a more reader friendly way.











References:

[1] Donald E. Knuth. Optimum binary search trees, Acta Informatica,  1971

[2] Kurt Mehlhorn. Nearly optimal binary search trees, Acta Informatica, 1975.